# Inflammation and Gastric Cancer

**DOI:** 10.3390/diseases10030035

**Published:** 2022-06-22

**Authors:** Aunchalee Jaroenlapnopparat, Khushboo Bhatia, Sahin Coban

**Affiliations:** 1Department of Internal Medicine, Mount Auburn Hospital, Cambridge, MA 02138, USA; ajoraenl@mah.harvard.edu (A.J.); khushboo.bhatia@mah.harvard.edu (K.B.); 2Department of Gastroenterology, Mount Auburn Hospital, Cambridge, MA 02138, USA

**Keywords:** inflammation, environmental factors, immunity, *H. pylori*, gastric microbiome, gastric cancer

## Abstract

Gastric cancer remains a major killer globally, although its incidence has declined over the past century. It is the fifth most common cancer and the third most common reason for cancer-related deaths worldwide. Gastric cancer is the outcome of a complex interaction between environmental, host genetic, and microbial factors. There is significant evidence supporting the association between chronic inflammation and the onset of cancer. This association is particularly robust for gastrointestinal cancers in which microbial pathogens are responsible for the chronic inflammation that can be a triggering factor for the onset of those cancers. *Helicobacter pylori* is the most prominent example since it is the most widespread infection, affecting nearly half of the world’s population. It is well-known to be responsible for inducing chronic gastric inflammation progressing to atrophy, metaplasia, dysplasia, and eventually, gastric cancer. This review provides an overview of the association of the factors playing a role in chronic inflammation; the bacterial characteristics which are responsible for the colonization, persistence in the stomach, and triggering of inflammation; the microbiome involved in the chronic inflammation process; and the host factors that have a role in determining whether gastritis progresses to gastric cancer. Understanding these interconnections may improve our ability to prevent gastric cancer development and enhance our understanding of existing cases.

## 1. Introduction

The development of cancer is complicated and involves both host and environmental factors. Microorganisms or chemical carcinogens can attack normal tissues and cause permanent DNA alteration. This phenomenon is known as the ‘subthreshold neoplastic state,’ the initial stage of cancer development. ‘Promotion’ is the second stimulus on the initiation sites when irritation or inflammation occurs. This dual-step stimulation promotes cell proliferation, recruits immune cells, and produces more reactive oxygen species to further damage DNA and reduce DNA repair mechanisms. Finally, the growth control system fails to function normally, and inflammation never subsides, resulting in tumor development [1,2]. Thus, chronic inflammation is strongly associated with cancer and a gateway to developing malignancy. Inflammation usually acts as a defense mechanism when the body faces insults, but it can also harm the body in certain circumstances.

For example, the development of atherosclerosis predisposes to severe conditions such as myocardial infarction and peripheral vascular disease. Inflammatory mediators attract immune cells to the site, communicate with resident and tumor cells, and induce an inflammatory response, promoting tumor growth. Inflammation promotes a suitable environment for tumor cells to grow, invade, and metastasize through the secretion of multiple inflammatory mediators. The perfect environment for tumor development from these immune responses is often referred to as the ‘tumor micro-environment’ (TME), which consists of immune cells, epithelial cells, and other cells and factors that provide a suitable environment for tumor growth. The tumor-promoting immunity outweighs the anti-tumor immunity in TME in the setting of inflammation-induced tumorigenesis [3].

Various mediators, including transcription factors, such as nuclear factor-kB (NF-kB), signal transducers, such as signal transducer and activator of transcription 3 (STAT3), proinflammatory cytokines, chemokines, and matrix metalloproteinases (MMPs), work together to enhance the tumor environment. The recruitment of inflammatory cells, such as macrophages and T-cells, in chronic inflammation promotes angiogenesis, malignant cell proliferation, and tumor invasion and even alters the efficacy of anti-tumor medication. Cancer can also secrete chemokines and communicate with stromal cells to recruit more inflammatory cells into the site. These immune cells can release reactive oxygen and nitrogen and induce inflammatory cytokines and growth factors that favor tumorigenesis [4]. Moreover, these oxidants can also cause DNA damage resulting in mutation (genetic alteration) or alteration of the presentation of genes without mutation (epigenetic alteration) [5]. Given this strong relationship between inflammation and cancers, chronic inflammatory markers are sometimes used as a predictor and assessment tool to determine cancer status, such as interleukin (IL)-6 and tumor necrosis factor-a (TNF-α), in hepatocellular carcinoma recurrence [6]. Therefore, more knowledge about inflammatory-related carcinogenesis mechanisms can benefit cancer prevention and identify treatment targets.

Inflammation is also associated with the transformation of cells from benign to malignant. Epithelial-mesenchymal transition (EMT) is a process in which normal epithelial cells transform themselves to acquire mesenchymal cell properties, such as invasion, apoptosis resistance, and metastasis. This process is observed in embryogenesis, wound healing, and various cancer types if dysregulation occurs [7]. Epithelial cells undergo multiple biochemical changes leading to morphological changes, abilities to migrate and invade, etc. EMT can even induce cancer stem cells to become chemo-resistant. Inflammatory mediators from chronic inflammation are involved with EMT by promoting its process. Transforming growth factor-b (TGF-β), TNF-α, hepatocyte growth factor (HGF), and hypoxia-inducible factor 1a (HIF1a) are the most relevant known mediators that facilitate EMT [8]. A clear example of this association is the Correa pathway, in which chronic gastric inflammation from *Helicobacter pylori* (*H. pylori*) infection-induced normal gastric epithelium develops into non-atrophic gastritis (Figure 1). The progression of gastritis can develop into gastric atrophy and metaplasia, which are precancerous lesions. These lesions eventually transform into low-grade dysplasia, high-grade dysplasia, and gastric cancer [9]. Detection of gastric cancer in its late stages is often related to poor prognosis; however, early detection of gastric cancer has a good to excellent prognosis, with a five-year survival rate of more than 90% after endoscopic resection. Therefore, in recent decades, research has been focused on improving endoscopic techniques which can provide early detection of precancerous lesions in the stomach.

## 2. Microenvironment Factors

Several environmental factors have been identified as risk factors for gastric cancer, such as salt-preserved foods, dietary nitrite, smoking, alcohol, GERD, *H. pylori* infection, Epstein-Barr virus infection, etc. (Table 1). These factors share the same mechanism for developing gastric cancer, which is chronic inflammation.

### 2.1. Autoimmune Gastritis

Autoimmune gastritis is a form of chronic gastritis resulting from attack by CD4+ T-cells on parietal cells in the stomach tissue. The infiltration of lymphocytes and plasma cells in the oxyntic mucosa, destroying parietal cells, leads to gastric tissue atrophy, which can later develop into gastric cancer through Correa cascades [10] (Figure 1). G-cell hyperplasia from the loss of the negative feedback loop due to loss of acid production from parietal cells leads to hypergastrinemia. Chronic elevation of gastrin can cause enterochromaffin (ECF) cell hyperplasia. ECF cell hyperplasia is a preneoplastic lesion of gastric carcinoid tumors [11].

### 2.2. Salt-Preserved Foods

From multiple ecological studies, the average salt intake in various populations has been found to be associated with gastric cancer mortality. A study from Japan found an association between urinary salt excretion and cumulative mortality from gastric cancer [12]. Another study in Mongolian gerbil models infected with *H. pylori* showed a relationship between inflammation caused by dietary salts and gastric cancer [13]. A dose-dependent association between dietary salt and inflammatory cell infiltration, serum gastrin, and *H. pylori* antibody titers has also been observed [14]. The mechanisms are still unclear, but several hypotheses have been proposed. First, dietary salts may alter gastric mucous viscosity and make it more sensitive to other carcinogens, such as N-nitroso compounds, leading to more severe inflammation and cancer development [15]. Second, sodium chloride ingestion causes increased S-phase cell numbers which are susceptible to mutation and abnormal cell development [16]. Third, dietary salts may enhance *H. pylori* colonization by increasing surface mucous mucin and decreasing gland mucous cell mucin [17]. Furthermore, dietary salt can also promote the CagA gene expression of *H. pylori* and its ability to translocate into the gastric epithelium [18]. All the above findings suggest that dietary salts contribute to an increased cancer rate by making the gastric mucosa susceptible to chronic inflammation.

### 2.3. Dietary Nitrites

Previous case-control and ecological studies have reported a positive association between dietary nitrites and the incidence of gastric cancer. Nitrate converts to nitrite and N-nitroso compounds which are carcinogens (nitrosation). These compounds are found in vegetables, drinking water, and processed meats. Advances in agricultural techniques have resulted in more nitrite accumulation in vegetables. Extensive fertilization can cause increased nitrate accumulation in drinking water [19]. Nitrosation results in free radicals that damage the gastric mucosa and cause DNA mutations. Ascorbic acid (Vitamin C) and beta-carotene are known to counteract nitrosation by eradicating these free radicals [20]. The inflammatory reaction arising from exposure to free radicals in the setting of nitrosation could explain the relationship between dietary nitrates and many cancer types. Previous studies have sought to determine the relationship between nitrite ingestion and gastric cancer. However, the data obtained have been inconsistent due to challenges in gathering information, as the cumulative intake of nitrites, and their compounds, cannot be accurately measured through saliva or urinary excretion. The quantity of nitrate intake has been obtained from subjects’ recall of food intake, increasing the recall bias. Multiple cohort studies have attempted to eliminate this recall bias, with most of these studies reporting a non-significant relationship between dietary nitrates and gastric cancer [21]. However, it is undeniable that nitrate compounds can promote carcinogenesis through the promotion of inflammation.

### 2.4. Gastric Surgery

Patients who undergo gastric surgery for other reasons, such as bariatric surgery, may have a higher risk of gastric cancer [22]. This could be explained by chronic inflammation from bile reflux in the stomach and decreased acid production in the stomach [23], especially in those patients who have undergone antrum resection, which is a critical location for gastrin secretion.

### 2.5. Alcohol

Alcohol is an independent risk factor for gastric cancer. The risk of gastric cancer increases five-fold in a person who simultaneously drinks more than 25 mg of alcohol and smokes over 20 cigarettes a day [24]. The mechanism is still unclear but could be explained by the solvent quality of alcohol, facilitating other carcinogens, such as tobacco carcinogens, to penetrate tissues more efficiently [25]. Aldehyde, an alcohol metabolic intermediate, is widely established to be a carcinogen in animals [26]. Moreover, some alcoholic beverages, such as hard liquor, contain nitrosamines, a known human carcinogen [27].

### 2.6. Smoking

A dose-dependent relationship exists between smoking and gastric cancer [28]. Tobacco contains more than 60 human carcinogens, including N-nitroso compounds [29]. In a study in mice, cigarette smoke components were found to bind covalently to DNA in lung and heart cells and to alter their function [30]. Similarly, smoking products can alter the DNA functioning of gastric cells and predispose them to cancer development.

### 2.7. Obesity

There are multiple hypotheses about the association between obesity and several cancers. Obesity alters insulin and insulin growth factor signaling, putting the body into a chronic inflammatory state and altering sex hormone metabolism. Obesity is associated with hyperinsulinemia and insulin resistance [31]. A study from Japan reported an association between circulating insulin levels and the development of gastric cancer [32]. Insulin stimulates insulin-like growth factor-1 (IGF-1) that can activate the PI3K/AKT/mTOR and Ras/Raf/MAPK pathways, increasing cellular proliferation [33]. Additionally, adipose tissue secretes inflammatory cytokines, such as IL-1β, IL-6, TNF-α, interferon-ϒ (IFN-Γ), and adipokines, such as leptin. Obesity strengthens Th1 cell response and thus dysregulates the balance of Th1/Th2 cell responses [34]. Obesity is also related to lower adiponectin levels, which have anti-tumor effects by increasing insulin sensitization and eliminating growth factors required for the development of tumor cells. On the other hand, leptin levels increase in obesity and stimulate proinflammatory cytokines. In some studies, leptin has been found to be positively related to colorectal cancer [35]. Lastly, excess adipose tissue increases estrogen production and reduces sex hormone-binding globulin, resulting in higher circulating endogenous estrogen, which increases the risk of cancer development [36].

### 2.8. Occupational Risk

Some occupations, such as those in agriculture, wood processing, coal mining, metal processing, and those which involve working with asbestos, are at higher risk of gastric cancer; these occupations are typically of low economic status [37]. Previous studies have concluded that the association observed could be confounded by the diet and lifestyles of people with low economic status [38]. The explanation for the higher gastric cancer risk could also be related to inflammation. Physical agents, such as asbestos and other mineral dusts, may act as co-carcinogens and cause direct irritation to the gastric mucosa [39]. The fertilizers used in agriculture contain N-nitroso compounds, which are carcinogens that can damage the DNA of the gastric epithelium by producing free radicals [40]. Chemical specks of dust from these compounds can damage stomach tissue by acting as carriers for other carcinogens [41].

## 3. Proinflammatory and Inflammatory Factors Involved in Chronic Inflammation Process in Gastric Cancer Pathway

### 3.1. Proinflammatory Cytokines

The main transcription factors that drive gastric cancer-related inflammation are NF-kB and STAT3. NF-kB is activated by the toll-like receptor (TLR)-MyD88 pathway, HIF-1a, IL-1a, and TNF-α. It is usually found in the early stage of cancer and can predict a good prognosis [42]. NF-kB induces various inflammatory cytokines, adhesion molecules, and cyclooxygenase (COX) 2 expression. Signaling through Gp130 receptors activates STAT3, which helps to maintain NF-kB activation. The dysregulation of Gp130 signaling causing constitutive activation is common in chronic gastric inflammation. Some cytokines that signal through Gp130, such as the IL-6 and IL-12 families, are associated with increased severity of gastric inflammation [43]. Mice with persistent activation of the Gp130 receptor, resulting from the mutation of the suppressor of the binding site of a cytokine to the receptor, rapidly developed gastric inflammation and dysplasia after *H. pylori* infection [44]. Higher STAT3 and STAT3 phosphorylation from persistent activation was associated with an unfavorable prognosis in gastric cancer [45]. This indicates that transcription factors can directly regulate tissue inflammation and tumorigenesis without involvement in immune cells.

Insults, such as infection with *H. pylori*, stimulate immune response through the TLR/MyD88 pathway and induce the COX-2/prostaglandin E (PGE) 2 pathway through NF-kB activation. The COX-2/PGE2 pathway induces IL-11, chemokine (C-X-C motif) ligand CXCL-1, CXCL-2, and CXCL-5, which help to promote tumor growth and maintain cancer cell stemness. The COX-2/PGE2 pathway function depends on the TLR/MyD88 pathway. These pathways need to work together to promote the tumor environment [46]. The nuclear receptor subfamily 4 group A member, 2 (NR4A2), is another transcription factor that NF-kB and PGE2 from the COX-2 pathway regulate. NR4A2 expression occurs in the chemoresistant stage and is associated with poor outcomes in patients receiving chemotherapy [47].

Host cytokines can suppress tumor activities by controlling the degree and type of inflammatory responses. Alternatively, cytokines can promote tumor growth and invasion when they are transformed by tumor cells. Inflammatory cytokines in gastric cancer-related inflammation pathways can be divided into three groups: cytokines that cause tumor progression, cytokines that activate anti-tumor activity, and cytokines that suppress anti-tumor activity.

Inflammatory cytokines that support gastric cancer-related inflammation include IL-1, IL-6, IL-18, TNF-α, and TGF-β. In normal circumstances, IL-1, which is mainly produced by macrophages, has an anti-tumor effect. However, in the setting of chronic inflammation, it can promote tumorigenesis. It can promote tumor adhesion, development, invasion, and metastasis through multiple pathways. IL-1 activates NF-kB and increases inflammatory factors with its subtypes, that include IL-1a and IL-1β. IL-1a promotes endothelial cell proliferation and angiogenesis and is, therefore, associated with liver metastasis. IL-1β promotes gastric cancer cell line proliferation via the tyrosine kinase pathway [48]. Moreover, IL-1β activates the NF-kB signaling pathway of myeloid-derived suppressor cells (MDSCs), which increases IL-6 and TNF-α secretion, enhancing tumor growth. In addition, IL-1 promotes the adhesion and metastasis of tumors by increasing vascular cell adhesion factor-1 expression [49]. IL-1β encourages tumor cell growth by increasing blood flow from angiogenesis by activating vascular endothelial growth factors (VEGF).

IL-6 is produced from various types of cells in TME under NF-kB regulation. IL-6 is linked to STAT3 phosphorylation through the JAK2/STAT3 pathway. Phosphorylated STAT3 enters the nucleus and induces oncogene and gene transcription factors, such as cFOX, TRF-1, and Bcl2. These processes promote tumor cell growth and differentiation, inhibition of apoptosis, angiogenesis, and adhesion. IL-6 also facilitates B-cell differentiation into plasma cells that can produce antibodies and enhances lymph node invasion and liver metastasis. Thus, elevated IL-6 levels are associated with more severe tumor stages, tumor invasion, and metastasis [50]. IL-18 is secreted from tumor-associated macrophages (TAMs) and stimulates tumor growth, similar to IL-6. Both IL-6 and IL-18 are predictors of prognosis in gastric cancer patients [51]. TNF-α promotes angiogenesis, tumor progression, and metastasis. Increased peritoneal carcinomatosis has been shown to occur in mice injected with TNF-α [52]. *H. pylori* produces high levels of TNF-α-inducing protein, which can bind to the cell surface and induce carcinogenesis of normal tissue through activation of NF-Kb [53].

TNF-α also upregulates the nitric oxide-dependent pathway and inhibits DNA repair. However, the predictive value of TNF-α levels for the severity of gastric cancer is controversial. Kabir et al. found that TNF-α was reduced in gastric cancer patients [54]. Forones et al. reported that, in stage III or IV gastric cancer, TNF-α levels were elevated in these advanced stages [55]. TGF-β has an inhibitory effect on the early stage of tumors by inhibiting the progression of cells from the G1 phase during proliferation. It also has a tumor-promoting effect when anti-proliferative signaling is disrupted by genetic alteration, leading to sustained proliferation [56].

Inflammatory cytokines that activate host anti-tumor activity include IFN-Γ, IL-12, and IL-18. IFN-γ is a Th1 type cytokine which has been shown to be associated with atrophy and SPEM in mice infected with *H. pylori*. IL-12 and IFN-Γ act via a positive feedback mechanism in relation to each other. IL-12 facilitates Th1 cell differentiation and increases IFN-Γ secretion. At the same time, IFN-Γ increases IL-12 secretion from phagocytes and dendritic cells [57]. IL-12 was found to act as a host anti-tumor immune response [58]. Inflammatory cytokines that suppress host anti-tumor activity include TGF-β and IL-10. IL-10 inhibits T-cell synthesis and the secretion of inflammatory cytokines from macrophages. The elevation of IL-10 levels is associated with poor prognosis [59]. TGF-β has immunosuppressive effects and helps the proliferation of regulatory T-cells (Tregs), resulting in the inhibition of immune responses from CD4+ and CD8+ T-cells [60].

With respect to other cytokines, IL-11 may be associated with gastric cancer, as treatment by IL-11 in mice resulted in chronic atrophic gastritis [61]. IL-17 may be negatively associated with gastric cancer as mice with IL17A developed less severe gastric inflammation [62]. IL-17A is secreted by CD4+ Th 17 cells, CD8+ T-cells, and natural killer (NK) cells and is associated with more severe disease [63]. IL22 is secreted by cancer-associated fibroblasts (CAFs). It promotes tumor invasion by signaling through the JAK/STAT pathway to promote gastric inflammation. IL22 receptor expression is also associated with lymphatic invasion [64]. IL-23 is one of the IL12 family that promotes gastritis through the Gp130 receptor pathway [65]. IL-32 can induce NF-kB activation and is associated with metastasis [66]. IL-33, a member of the IL-1 family, is a cytokine produced from epithelial cells that can enhance Th2 expression of cytokines, such as IL-5 and IL-9, and induce inflammation in the stomach. IL-33 uses an IL-13-independent mechanism to promote SPEM, a precancerous gastric cancer lesion [67].

Different types of cytokine production result in markedly different consequences. C57BL/9 mice with a predominantly Th1 immune response are more susceptible to gastric cancer after being infected with *H. pylori*. IFN-Γ, a cytokine involved in the Th1 immune response, can cause gastritis when injected into mice. On the other hand, BALB/c mice with an immune response skewed towards Th2 were observed to eradicate more bacteria and to develop less severe gastric inflammation. IL-4, a cytokine involved in the Th2 immune response, can prevent gastric inflammation in mice [68]. Interestingly, concurrent infection with helminths can decrease the level of gastric inflammation by skewing toward Th2 over Th1 immune responses, which might explain the African enigma [69].

Chemokines are the inflammatory mediators that attracts leukocytes to the inflammatory areas when they are under the influence of proinflammatory cytokines. They communicate with leukocytes using G-protein-coupled receptors. Overexpression of stromal-derived-factor (SDF)-1, CCL7, and CCL21 are associated with lymph node invasion [70]. The CXC chemokine receptor 4 (CXCR4) works with SDF-1 and is associated with a poor prognosis. CCR3, CCR4, CCR5, and CCR7 are also markers of poor outcome [71]. IL-8 is a chemokine produced by macrophages, endothelial cells, epithelial cells, and fibroblasts. It attracts neutrophils and increases the risk of atrophic gastritis and gastric cancer. IL-8 binds to its receptor CXCL1/CXCL2 and promotes angiogenesis, proliferation, and the invasion of tumor cells. MMPs are other inflammatory mediators that are usually expressed in the context of *H. pylori* infection and different kinds of inflammation. MMPs release chemoattractants to recruit more inflammatory cells and promote tumor progression. Some types of MMP, such as MMP3, and MMP7, can predict worse survival outcomes in patients infected with *H. pylori* [72].

### 3.2. Immune Cell Types Playing a Role in the Inflammatory Cascades

Many types of immune cells are involved in the inflammatory responses in gastric cancer. These immune cells are recruited to reside in the TME and adjust the environment such that it favors tumor growth, invasion, and metastasis. In response to tissue injury, leukocytes, such as neutrophils and monocytes, gather from the venous system to the injured sites attracted by cytokines released from resident immune cells and tumor cells. For instance, neutrophils are attracted by cytokines, such as IL-1, TNF-α, and chemokines, such as IL-8. They travel to the site through the endothelium using adhesion molecules such as E-selectin. The vascular cell adhesion molecule-1 facilitates immobilization of neutrophils at the injury site and transmigration to the site through endothelial walls by MMPs. Monocytes migrate to the area after being attracted to IL-1β, TNF-α, and chemokines, such as the chemokine ligand CCL2. Monocytes differentiate into mature cells and act as the primary producer of multiple inflammatory cytokines, such as TGF-β, IL-1, and IGF-1, which promote cancer cell survival [73]. They also favor tumorigenesis by modulating tumor cell growth and angiogenesis. Fibroblasts stimulated by TGF-β will follow to the site and promote tissue proliferation and remodeling.

The persistent inflammation from chronic *H. pylori* infection, which results in immune cell migration and cytokine secretion that finally turns inflammatory tissue into a pre-malignant state, represents a perfect example of a TME. The gastric cancer TME is enriched in MDSCs, Tregs, and TAMs. TAMs are derived from monocytes in the venous system and are a poor prognostic indicator of tumors. TAMs have both anti-tumor and tumor-promoting roles. Monocytes recruited to the TME will turn into non-polarized (M0) macrophages that can later polarize into two distinct types: M1, or classically activated, macrophages are involved in acute inflammatory responses and are responsible for an anti-tumor effect, and M2, or alternatively activated, macrophages are involved in tumor progression [74].

M2 macrophages, the most expressed TAMs in the TME, are stimulated by Th2 cytokines, such as IL-4, IL-10, and IL-13. M2 macrophages act to promote tissue remodeling and tumor progression. Tumor and stromal cells can directly communicate with and recruit monocytes to turn into TAMs, secrete CSF-1 and IL-4 and induce M2 polarization [75]. TAMs themselves secrete IL-6, IL-8, and IL-10 that promote tumor growth. TAMs also secrete VEGF, TNF, IL-1, IL-8, PDGF, and FGF, promoting angiogenesis [76]. Moreover, TAMs promote tumor invasiveness by activation of the EMT factor. TAMs can also mobilize to peripheral blood, reside in the pre-metastatic niches, and facilitate tumor activities. MDSCs include various types of immature cells of myeloid origin that can suppress T-cell responses. Their numbers increase rapidly in the setting of inflammation and are associated with advanced gastric cancer [77]. All activated myeloid cells produce reactive oxygen species (ROS) and reactive nitrogen species. These reactive agents inhibit the migration of CD8+ T-cells, thus reducing anti-tumor effects [78].

Tumor-infiltrating lymphocytes are a group of lymphocytes found in the TME. They consist of T-cells, B-cells, and natural killer (NK) cells. There are many types of T-cells, including CD8+ cytotoxic T-cells, CD4+ T-helper cells, FOXP3+ regulatory T-cells, CD45RO memory T-cells, and NK cells. These lymphocytes act against tumor cells, promote tumor cell apoptosis, and are associated with better prognosis [79]. However, CD8+ T-cells can produce IL-17, which can promote inflammation and is related to poor outcomes [80]. Th1 cells are subtypes of CD4+ T-helper cells that can produce IFN-Γ and activate CD8+ cytotoxic T-cells. Th2 cells are another subtype of CD4+ T-helper cell that can produce IL-4 and are involved in humoral immunity. Th1 cells are more effective than Th2 in inducing anti-tumor effects. A high Th1/Th2 ratio can predict favorable outcomes in gastric cancer patients [81]. Th17 and Th22 are associated with tumor progression [82]. FOXP3+ regulatory T-cells are associated with poor outcomes [83]. NK cells consist of CD56+ and CD57+ subtypes. CD57+ is more related to poor outcome [84]. B-cells, such as CD19+ and CD20+, appear to be associated with favorable outcomes [85].

CAFs are another type of cell in the gastric TME. CAFs are activated from the interaction between cancers and stromal tissue and are derived from cells such as normal fibroblasts and bone marrow mesenchymal stem cells (MSCs). They produce TGF-β, bFGF, platelet-derived growth factor (PDGF), and VEGF to regulate tumor growth [86]. They secrete IL-6, which induces an inflammatory response after *H. pylori* infection. CAFs release multiple proinflammatory factors that promote EMT, such as COX2, CXCL-1, CXCL-9, CXCL-10, CXCL-12, and fibroblast-specific protein-1 [87].

MSCs can differentiate into CAFs and tumor-associated MSCs after they migrate to tumors and can promote EMT. MSCs mainly promote angiogenesis through fibroblast growth factor (FGF), PDGF, and VEGF secretion [88].

### 3.3. Genetic Alterations Accumulated in Inflamed Epithelial Cells in Gastric Carcinoma

Host genetic factors also play a large part in gastric cancer expression and the spectrum of disease. The Cancer Genome Atlas research network classifies gastric cancer into four subtypes. The first type is the EBV-positive subtype which involves mutation of PIK3CA, DNA hypermethylation, JAK2 amplification, PD-L1, and PD-L2 amplification. The second type is the microsatellite instability subtype, which involves hypermutation and epigenetic silencing of the mismatch-repair gene MLH1. The third subtype is the chromosomal instability subtype, which exhibits TP53 mutations, aneuploidy, and focal amplification. This subtype is related to the intestinal type of gastric cancer. And the last subtype, the genomically stable subtype, involves alteration of CDH1 or RHO family genes [89].

These genetic changes usually occur at the early stage of gastric cancer [90]. Chronic inflammation from *H. pylori* infection and other insults are associated with an increased inflammatory response that can cause genetic alteration in hosts, which predisposes to carcinogenesis. C:G to T:A transitions are associated with alteration of the spontaneous deamination process, which can cause mutations. This transition is found in aging and is the most commonly detected transition in gastric cancer patients [91]. C:G to A:T transversion is the second most common genomic change in gastric cancer and is associated with smoking and ROS [92].

As the inflammation process is closely related to gastric cancer development, genetic alteration in genes involved with inflammatory mediators can increase gastric cancer. Specific genetic alterations that affect the inflammatory cytokine response can increase gastric cancer risk. For example, for the IL-1 gene, persons with homozygous IL-1RN*2 alleles, carriers of IL-1β-511T allele carriers, and of the IL-1RN*22 allele, are more susceptible to developing gastric cancer [93]. For TNF-α, persons with the 308*A allele polymorphism or the −863C/A polymorphism are at higher risk. But TNF-α −308G/A and −1031 T/C polymorphisms have protective effects against *H. pylori* infection [94]. Polymorphisms in the IL-10 gene (−1082A/G, −819T/C; −592 A/C) are associated with gastric cancer risk [95]. The genetic alterations associated with cancer risk can also occur in other inflammatory cytokines, such as IL-8 and IL-17.

Genetic alteration can occur at any point in inflammatory cascades, such as in the TLR and nod-like receptor. For example, TLR4 polymorphisms, such as +896A/G and +1196C/T, are associated with gastric cancer [96]. Oxidative stress of inflammatory reactions, such as by ROS, can cause DNA damage. Mice with DNA repair enzyme defects had a higher risk of DNA damage from oxidative stress. Mice with p53 tumor suppressor gene inactivation were observed to be at higher risk of gastric hyperplasia and mutation [97]. Polymorphisms in other mismatch repair genes, such as ERCC2 and XRCC1, are associated with gastric neoplasm [98]. COX2 gene alteration was also found to be associated with gastric cancer development (1195AA polymorphism) [99].

## 4. *H. pylori* Infection

*H. pylori* infection plays a crucial role in gastric carcinogenesis. *H. pylori* is responsible for at least 89% of all non-cardia gastric malignancies [100]. Given the importance of *H. pylori* in the development of gastric cancer, the International Agency for Research on Cancer identified chronic *H. pylori* infection, which is acquired as early as infancy, as a primary cause of gastric adenocarcinoma in 1994 [101]. *H. pylori* infection, if left untreated, can trigger a slow long-term carcinogenic process, leading to antral-predominant chronic active gastritis and, ultimately, multifocal atrophic gastritis, which is a risk factor for intestinal and, less often, diffuse gastric adenocarcinomas [102]. However, intriguingly, only a small percentage of people infected with *H. pylori* develop gastric cancer. Therefore, many factors, including genetic susceptibility, environmental factors, and *H. pylori* bacterial strain differences, all play a role in facilitating the neoplastic process in chronic *H. pylori* infection.

### 4.1. Role of Bacterial Virulence Factors in Chronic H. pylori Infection and the Pathway of Developing Gastric Cancer

*H. pylori*-induced tissue injury is initiated by bacterial attachment and the subsequent release of enzymes and other microbial products that cause cellular damage (Figure 2).

#### 4.1.1. Motility and Colonization

*H. pylori* bacteria have unique mechanisms to resist the extreme acidic conditions of the stomach. They travel using flagellar motility, which permits the bacteria to enter the mucus layer. Several studies have reported that mutations in genes encoding specific flagellar proteins, such as fliD, FlaA, and FlaB, impede *H. pylori* motility, diminishing or preventing the bacteria from colonizing the stomach mucosal layer [103,104,105]. *H. pylori* motility is also influenced by chemotactic activity in response to various molecules, such as mucin, sodium bicarbonate, urea, sodium chloride, and certain amino acids [106,107]. Several *H. pylori* genes involved in chemotactic stimuli reception, signal transduction, and processing have been discovered during infection [108]. Chemoreceptors, including T1pA, B, C, and D, a CheA kinase, a CheY responsive regulator, and various coupling proteins are essential for its chemotaxis [109].

#### 4.1.2. Bacterial Attachment

*H. pylori* prefers gastric epithelial cells. The strong adherence of *H. pylori* to the gastric cell surface was confirmed by electron microscopy, which revealed membrane attachment pedestals similar to those seen in enteropathogenic *Escherichia coli* [110,111]. Adhesins and outer membrane proteins mediate their attachment to the gastric cells. Bacterial adhesins must detect and attach precisely to host receptors expressed on the cell surface for this process to take place [112]. This attachment process can potentially change the epithelial cell’s morphology or function or activate specific bacterial functions, making them more hazardous. Bacterial membrane proteins encoded by genes located in the cytotoxin-associated gene (cag) pathogenicity island form channels in the epithelial cell membrane, allowing bacterial factors to access the cytoplasm directly [113]. BabA (HopS), OipA (HopH), and SabA (HopP) are three Hop proteins, which are adhesin proteins implicated in the pathophysiology of *H. pylori* infection [114]. BabA, the most well-studied of these three adhesin proteins, facilitates host cell binding to fucosylated Lewis b (Le(b)) blood group antigens [115]. Although OipA functions as an adhesin, it promotes inflammation by boosting IL-8 expression [116]. SabA is involved in the binding to sialic acid-containing glycoconjugates [117]. The function of Lewis antigen expression in bacterial attachment is unclear. However, the simple replacement of non-sialylated Lewis antigens by sialylated Le(x) or Le(a) has been associated with *H. pylori*-induced gastric inflammation and cancer [118,119].

#### 4.1.3. Release of Enzymes

Urease catalyzes the hydrolysis of urea to carbon dioxide and ammonia, which buffer and attenuate the acidity of the stomach environment, ultimately aiding *H. pylori* in colonization [120]. Hydrogenase is part of a signaling cascade that triggers the formation of an alternate pathway, allowing *H. pylori* to utilize molecular hydrogen as a source of energy for its metabolism [121].

#### 4.1.4. Nickel

Nickel is essential for *H. pylori* because it serves as a cofactor for two key enzymes, urease, and hydrogenase, that play an indispensable role. Nickel binds at the N-terminal NHE motif of the Ni-metallochaperone HypA, which is essential in the maturation of urease and hydrogenase, critical for the bacteria’s acid survival [122]. In addition, nickel-dependent urease activation requires NiuBDE, a unique *H. pylori* nickel transport system [123]. Furthermore, two histidine-rich paralogous nickel-binding proteins, Hpn and Hpn-2, are required to maintain a nontoxic intracellular level of nickel and, in turn, for gastric colonization [124].

#### 4.1.5. Cag Pathogenicity Island

The cag pathogenicity island (cagPAI) gene is intimately linked to *H. pylori* virulence as it encodes for a T4SS (type IV secretion system) and the bacterial oncoprotein, the cytotoxin-associated gene A (CagA) [124,125]. Bacterial membrane proteins encoded by genes found in the cagPAI create channels in the epithelial cell membrane at the adhesion site, allowing bacteria to interact directly with the cytoplasm.

#### 4.1.6. CagA and Vacuolating Cytotoxin A (VacA)

CagA and vacuolating cytotoxin A (VacA) are two *H. pylori* virulence factors that vary between strains which appear to impact the bacteria’s pathogenicity and the risk of gastric precancerous lesions and adenocarcinoma [125]. Higher levels of mucosal inflammation and advancement of precancerous lesions are linked to differences in these virulence factors [126,127].

#### 4.1.7. Type 4 Secretion System

Virulent strains of *H. pylori* encode cagPAI, a gene that expresses a type IV secretion system (T4SS). The T4SS generates a syringe-like pilus structure for injection and translocation of virulent components into host target cells, such as the CagA effector protein and peptidoglycans. This process includes several T4SS proteins, such as CagI, CagL, CagY, and CagA, attaching to a host cell integrin before delivering CagA across the host cell membrane [128,129].

CagA is a hydrophilic, surface-exposed bacterial protein produced by most clinical isolates of *H. pylori*; it is not cytotoxic, but antigenic and serologically detectable [130]. It causes specific morphological changes in epithelial cells by altering the cell polarity, resulting in the elongation of cells called the “hummingbird” phenotype [131,132]. Therefore, CagA causes changes in the cytoskeleton that are linked to developing gastric cancer. Once inside the host cell through T4SS, CagA is phosphorylated by oncogenic tyrosine kinases and mimics host cell factors to activate or inactivate intracellular signaling pathways [31,133,134,135]. CagA undergoes tyrosine phosphorylation at a Glu-Pro-Ile-Tyr-Ala (EPIYA) motif, a variable C-terminal CagA region that can be combined by different EPIYA segments (EPIYA-A, EPIYA-B, EPIYA-C, and EPIYA-D) [136]. Most cagA (+) *H. pylori* strains contain EPIYA-A and EPIYA-B segments, whereas EPIYA-C and EPIYA-D segments are associated with Western and Eastern strains, respectively [137]. *H. pylori* strains bearing EPIYA-D, or at least two EPIYA-C segments, in their cagA gene have an increased risk of developing cancer [138]. Furthermore, a Brazilian study found that first-degree relatives of gastric cancer patients are more likely to be infected by *H. pylori* strains carrying two or more EPIYA-C segments [139]. The phosphorylation allows binding to SH2 domain-containing proteins, such as SHP2 tyrosine phosphatase, promoting host cell alterations [140]. In vitro and in vivo studies have shown that the Epstein-Barr virus (EBV) can cause SHP1 promoter hypermethylation, indicating a link between EBV and *H. pylori* coinfection and developing gastric cancer [141]. Krish et al. used B-cell chronic lymphocytic leukemia-derived cells as an infection model where they reported that CagA was associated with *H. pylori*-induced gastric MALT lymphoma and that infiltrating lymphocytes in the stomach mucosa could interact directly with *H. pylori* [142]. According to the researchers, CagA was immediately injected and tyrosine phosphorylated by cell kinases from the Src and Abl families. This suggests that inhibitors of these kinases might effectively treat MALT lymphoma caused by *H. pylori*.

Another virulence factor of *H. pylori* strains is VacA, which functions as a passive urea transporter that has the potential to increase the permeability of the gastric epithelium to urea, hence facilitating *H. pylori* infection [143]. VacA virulence is dependent on the tyrosine phosphatase receptor function in gastric epithelial cells [144]. *H. pylori* strains bearing different VacA alleles have differing toxicity [145]. The VacA gene is found in all *H. pylori* strains, but only those that encode the cagPAI gene express VacA [146].

VacA and CagA-producing strains generate more intense tissue inflammation and cytokine production [147,148]. CagA (+) strains are found in 85–100% of individuals with duodenal ulcers, compared to 30 to 60% of infected patients who do not develop ulcers [149]. Moreover, CagA strains have been linked to an increased risk of precancerous lesions and gastric cancer [150]. Specific amino acid sequences in the CagA protein may be linked to the risk of cancer [39]. Many of the virulence factors listed can coexist in the same *H. pylori* strain, making it difficult to determine the most relevant ones. Furthermore, CagA expression is linked to gastric cancer and duodenal ulcer, although these two diseases seldom, if ever, coincide, limiting the importance of this virulence factor in disease processes.

### 4.2. Inflammatory Response

*H. pylori* infection triggers various innate and adaptive immunological responses in the host [151,152]. Various *H. pylori* antigens, such as lipoteichoic acid, lipoproteins, and lipopolysaccharide, bind to gastric cell receptors, including TLRs, found on epithelial cell membranes and intracellular vesicles [153,154]. This promotes NF-kB and c-jun N-terminal kinase activation, among other signaling pathways, leading to proinflammatory cytokine release [155]. In addition, CagA injection through T4SS causes cytokine generation, another NF-B-dependent process [156]. Consequently, neutrophils and mononuclear cells infiltrate the gastric mucosa and produce nitric oxide and ROS [157]. Adaptive immunity also has a role, especially in CD4+ and CD8+ T-cells involved with preferential activation of CD4+ cells [158]. Studies have revealed a Th1-polarized response in *H. pylori*-positive patients, characterized by low levels of IL-4 (a Th2 cytokine) and increased gamma interferon, tumor necrosis factor, and interleukins, including IL-1, IL-6, IL-7, IL-8, IL-10, and IL-18. Upregulated cytokines may boost proinflammatory effects during *H. pylori* infection, except for IL-10, which appears to play a role in limiting the inflammatory response [159,160]. *H. pylori*-specific serum IgM antibodies can be observed in patient serum four weeks after infection [161]. In chronic infection, serum IgA and IgG immunoglobulins canalize toward many bacterial antigens [162,163]. This inflammation is asymptomatic in most infected patients, but it raises the risk of duodenal and gastric ulcer disease and developing gastric cancer in the long term [164].

### 4.3. Host Factors That Affect Developing Gastric Cancer

Polymorphisms in genes encoding IL-1β, the IL-1 receptor antagonist, TNF-α, and IL-10 have been linked to the initiation and modulation of the inflammatory response and have been associated with susceptibility to carcinogenesis [165,166].

#### 4.3.1. EBV

Although the role of EBV in gastric carcinogenesis, directly or indirectly, is contested, it is estimated that between five and ten per cent of gastric cancers globally are linked to EBV [167,168]. EBV-associated gastric cancers were found to have distinct clinicopathologic characteristics. Male predominance, location in the gastric cardia or postsurgical gastric stump, lymphocytic infiltration, a lower frequency of lymph node metastasis, a diffuse type of histology, and potentially a better prognosis, are all features of gastric cancer associated with EBV [169,170]. These are suitable for immune checkpoint inhibitor treatment, due to the overexpression/amplification of programmed cell death ligand-1 in these tumors.

#### 4.3.2. Familial and Genetic Factors

Gastric cancer is roughly three times more common among first-degree relatives of individuals with the disease than in the general population [171]. This could be explained by the fact that *H. pylori* infection runs in families and household contacts, and the potential significance of inherited IL-1 gene polymorphisms in the inflammatory response to *H. pylori* and its association with gastric cancer [69,70].

#### 4.3.3. Environmental Factors

Factors that seem to have a role in the development of gastric cancer, mainly when *H. pylori* infection is present, include a high salt diet, low iron, smoking, and bile reflux after surgery.

#### 4.3.4. Other Gastric Pathology

It is well known that gastric dysplasia and intestinal-type adenocarcinoma are more common in people with pernicious anemia and autoimmune gastritis. Both neoplastic and non-neoplastic gastric epithelial polyps have been linked to the development of gastric cancer [172].

### 4.4. H. pylori: Mechanism of Disease

#### 4.4.1. *H. pylori*-Mediated Autophagy and Precancerous Lesions

*H. pylori* infection typically occurs in childhood or early adolescence, and gastric cancers are usually not clinically recognized until four or more decades later. During this significant latency period, a precancerous process occurs. Sequential histopathologic stages are chronic active non-atrophic gastritis, multifocal atrophic gastritis, intestinal metaplasia, dysplasia, and invasive carcinoma [173,174] (Figure 3). Population-based landmark studies from two Scandinavian nations showed the increased risk of gastric cancer in individuals with these premalignant lesions [175,176].

#### 4.4.2. The Preneoplastic Cascade

Non-atrophic gastritis predominately occurring in the antrum, is characterized by dense, band-like infiltration of lymphocytes, macrophages, and plasma cells. This phase is the initial stage of chronic *H. pylori* infection and is the most common histologic abnormality in people who have an *H. pylori*-related duodenal peptic ulcer. Notably, it is not linked to elevated cancer risk. Only in a subset of patients does atrophy with intestinal metaplasia develop, starting a cascade that may turn into adenocarcinoma. The rate of progression from non-atrophic gastritis to atrophic gastritis/intestinal neoplasia in large cohorts was between 0.1 and 0.9% [177].

Atrophic gastritis is marked by multifocal loss of the original gastric glands and is the first histopathologic lesion of the preneoplastic *H. pylori* cascade. Atrophic gastritis may be followed by the development of glands presenting an intestinal phenotype referred to as intestinal metaplasia. The neutral-pH gastric mucins are replaced by acid mucins, which can be sialic or sulfated [178,179]. Gastric intestinal metaplasia is classified into the following kinds, based on the types of mucins expressed and immunohistochemistry: Type I (complete) intestinal metaplasia, Type II (incomplete) intestinal metaplasia, and Type III (incomplete) intestinal metaplasia. Complete-type intestinal metaplasia comprises goblet cells and absorptive cells with a brush border. It has reduced or absent expression of gastric mucins with expression of MUC2, intestinal mucin, instead [180]. However, incomplete intestinal metaplasia is characterized by goblet and columnar nonabsorptive cells without a brush border and co-expression of gastric mucins and MUC2.

The earliest metaplastic glands observed in chronic *H. pylori* infection morphologically resemble those of the small intestine, with eosinophilic absorptive enterocytes with a brush border alternating with mucus-producing goblet cells. The loss of acid-secreting parietal cells is assumed to cause these alterations. In more advanced stages, the glands become bordered by irregular goblet cells resembling the colonic phenotype [181]. A systematic review of ten observational studies found that individuals with incomplete metaplasia had a four- to eleven-fold greater risk of gastric cancer than those with complete metaplasia or absence of incomplete type metaplasia [182,183]. Incomplete metaplasia is commonly seen among early gastric adenocarcinomas. Colonic metaplasia is thought by some to be an early stage of dysplasia, necessitating more frequent endoscopic monitoring than small intestine metaplasia [184]. Metaplasia initially appears at the antrum-corpus mucosa junction, particularly in the incisura angularis, the site of the loss of acid-secreting parietal cells and oxyntic atrophy. Metaplasia foci become larger and more abundant over time, expanding to the antrum and gastric corpus. Understandably, cancer risk increases with the size of the atrophic and metaplastic region. In over one per cent of people infected with *H. pylori*, corpus-predominant gastritis develops, leading to multifocal gastric atrophy, intestinal metaplasia, and hypo- or achlorhydria. Furthermore, with the elevation in gastric pH, a change in the gastric flora occurs, with colonization by anaerobic bacteria, forming carcinogenic nitrosamines [185]. Hence, hypochlorhydria, low levels of pepsinogen I, and gastrin in the serum can be utilized as indicators of gastric atrophy and cancer risk [186].

#### 4.4.3. Spasmolytic Polypeptide-Expressing Metaplasia Pathway

Another pathway to gastric neoplasia is called “spasmolytic polypeptide-expressing metaplasia” (SPEM), which is associated with oxyntic mucosa atrophy and significant expression of the trefoil factor family 2 protein (TFF2, previously known as a spasmolytic polypeptide). SPEM is also known as pseudopyloric metaplasia, and was found to have a significant association with chronic *H. pylori* infection and gastric cancer [187,188,189] (Figure 1 and Figure 3).

SPEM has been demonstrated to arise after oxyntic atrophy in animal models by differentiation of zymogen-secreting chief cells into cells with a mucous secretory profile [190]. Furthermore, inflammatory cytokines may have a role in chief cell reprogramming, considering that SPEM is a possible physiologic repair mechanism for recruiting reparative progenitor cells in response to mucosal lesions [191]. *H. pylori* may access deeper locations when SPEM is present, which may allow it to propagate and evolve into a hyperproliferative condition, rendering the infected and inflamed mucosa more vulnerable to the formation of deleterious mutations in the stem or progenitor cell populations [192].

Gastric dysplasia is a direct precursor of gastric adenocarcinoma, which is the final stage of the cascade. It is characterized by neoplastic cytologic and architectural features that are confined to the basement membrane. On endoscopy, lesions might appear as flat, polypoid, or depressed. Gastric dysplasia is similar to gastric cancer, with regional variation seen and male predominance. Patients are about a decade younger than those with gastric cancer, with an average age of 61 years versus 70 years [193].

## 5. The Gastric Microbiome and Effects of *H. pylori* Infection on the Gastric Microbiome

The discovery of *H. pylori* in the stomach of patients with gastritis and peptic ulcers has challenged the idea that the stomach is a sterile organ due to its strongly acidic environment [194]. Classical methods for studying the human gastric microbiome comprise microbiologic techniques, including culture, isolation, and identification. However, under these standard culture conditions, only a small number of gastric microorganisms can be grown; thus, most microorganisms cannot be identified by this approach. Some microorganisms, such as Lactobacillus, Veillonella, and Clostridium spp. Can be isolated from the human stomach by culture-dependent methods [195]. Furthermore, with newer techniques, such as microarrays, random shotgun sequencing, and next-generation sequencing, a large number of taxa have been detected. The microbial load of the stomach is much lower than that of the intestine (approximately 10^2^–10^4^ colony−forming units (CFU)/mL versus 10^10^–10^12^ CFU/mL) [196]. Intriguingly, human gastric juice is distinct from the gastric mucosa in terms of microbial load. Gastric juice has a diverse microbial community, which is mostly dominated by Firmicutes, Actinobacteria, and Bacteroidetes, whereas gastric mucosa mainly includes Proteobacteria and Firmicutes [197,198]. Moreover, bacteria presenting in the oral cavity and distal locations of the stomach, including Clostridium, Veillonella, and Lactobacillus, can transiently colonize the stomach as well [199]. Therefore, the microbial load in gastric juice might not represent the gastric microbiome all the time.

The mechanisms contributing to inter-individual variations in the gastric microbiome community are not well understood. Many factors, including age, sex, ethnicity, childbirth delivery mode, diet, lifestyle, geography, use of antibiotics, proton pump inhibitors (PPI) or histamine H2 receptor antagonists, and the presence of *H. pylori*, affect the gastric microbiota composition [200] (Figure 4). In a healthy stomach, the acidic environment prevents the overgrowth of bacteria and reduces the risk of infection [201], and, conversely, reducing gastric acid by long-term therapy with PPI or H2 antagonists can lead to bacterial overgrowth [202].

The relationship between *H. pylori* and stomach-resident bacteria is not fully known. In the stomach of *H. pylori*-infected patients, *H. pylori* is the predominant bacterium [203]. However, some studies have reported that low numbers of *H. pylori* were identified by broad-range polymerase chain reaction and 16S rDNA sequence analysis in patients who were negative for *H. pylori* infection by traditional methods, including rapid urease test, serologic analysis, histopathology, and culture. Therefore, a cutoff value for *H. pylori* infection by pyrosequencing has been used to define its presence in human gastric samples [204]. It has been found that alterations in gastric microbiome composition can increase the risk of developing gastric cancer [205].

*H. pylori* can modulate the gastric environment, altering the environment of resident microorganisms [206]. In a study from Sweden, Andersson et al., revealed that individuals who were *H. pylori*-negative had a more diverse gastric microbiome than those testing positive [207]. Thus, *H. pylori* can potentially set the stage for gastric cancer in this way. In addition, it was found that the eradication of *H. pylori* increased microbial diversity in the stomach. In recent studies, bacterial overgrowth in the stomach has been reported in various precancerous conditions. Li et al. demonstrated that different histologic stages of gastric carcinogenesis, from gastritis to carcinoma, had an inverse relationship with *H. pylori* quantity and microbial diversity in non-cancer gastric biopsy samples [208].

*H. pylori* infection is a crucial cause of gastric mucosal damage. It can cause a chronic inflammatory process in the gastric epithelium, leading to changes in cellular kinetics, which results in atrophy of the glands. IM, an adaptive response to a foreign environment, plays a role in the inflammatory response. *H. pylori* infection could also trigger the expression of inducible nitric oxide synthase, producing many nitric oxides and N-nitroso compounds, which induce DNA damage in the gastric epithelial cells [209,210]. Bacteria such as Veillonella, Clostridium, Haemophilus, and Staphylococcus can contribute to forming N-nitroso compounds [211], which play an essential role in developing gastric cancer [212,213] (Figure 4).

## 6. Conclusions

Screening strategies in gastric cancer are currently performed based on endoscopic techniques. Identifying higher-risk patient groups combined with reliable endoscopic diagnosis and accurate assessment of the chronically inflamed stomach is vital to the early detection and successful treatment of gastric cancer. Chronic inflammation in gastric cancer development involves many complex factors and processes. Therefore, it is crucial to define a multi-omics strategy in studying chronic inflammation and gastric cancer to thoroughly understand the molecular mechanisms of chronic inflammation in cancer development. Discovering more reliable and effective biomarkers targeting the chronic inflammation process associated with cancer to predict the development of cancer in the very early stages can effectively prevent cancer and enable the design of a reasonable assessment index to evaluate the effects of cancer prevention interventions. Future research should focus on the feasibility and reproducibility of combining multi-omics strategies with an endoscopy-led staging algorithm for premalignant gastric lesions. Such an approach would help detect premalignant gastric lesions, allow more accurate risk assessment, and even avoid unnecessary tissue biopsies leading to a reduced burden on patients under surveillance and on pathology services.

## Figures and Tables

**Figure 1 diseases-10-00035-f001:**
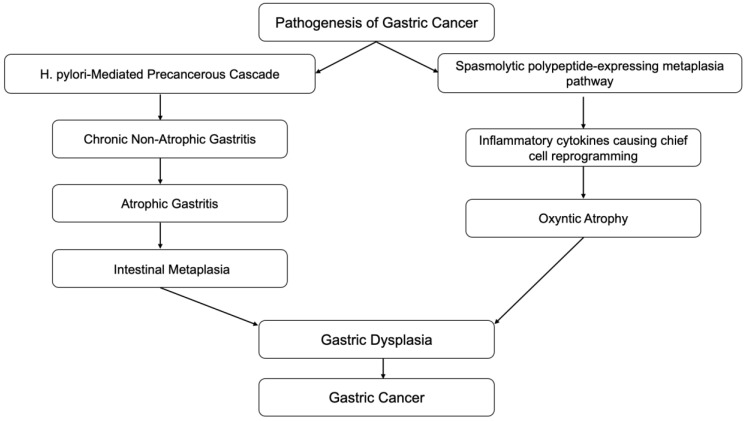
Correa cascades.

**Figure 2 diseases-10-00035-f002:**
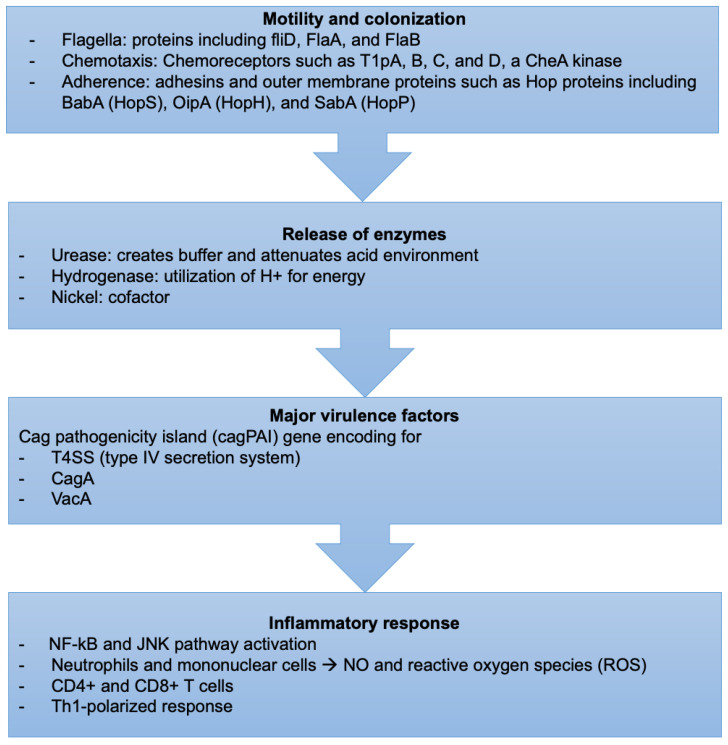
Sequence of events describing the virulence factors and mechanism of *H. pylori* infection.

**Figure 3 diseases-10-00035-f003:**
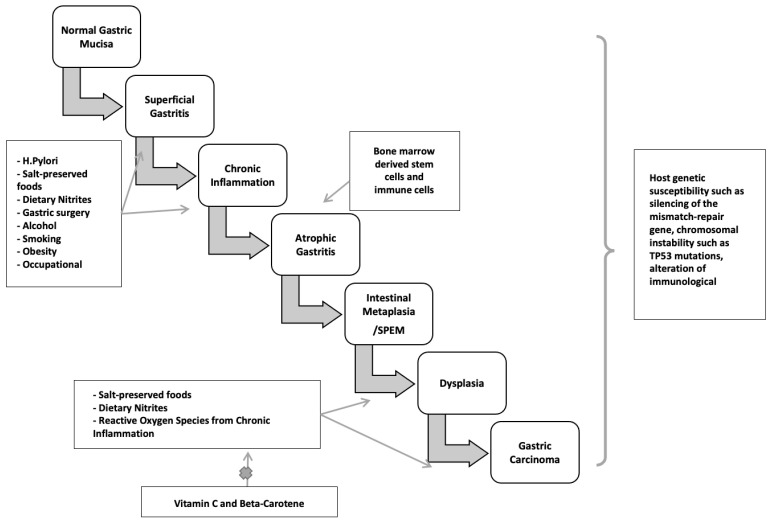
Two major pathogenetic pathways to gastric cancer.

**Figure 4 diseases-10-00035-f004:**
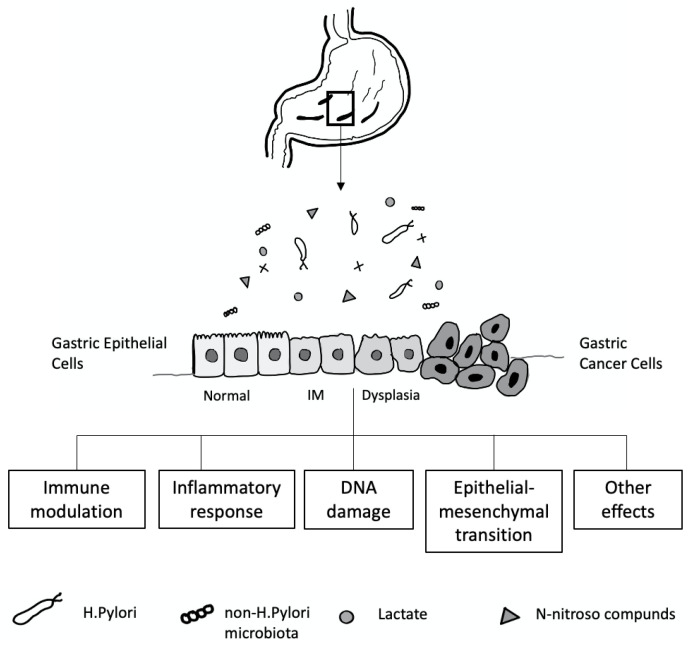
Gastric microbiota and the development of gastric cancer cell relationship.

**Table 1 diseases-10-00035-t001:** Environmental factors related to the development of gastric cancer.

Environmental Factors	Possible Mechanisms
*H. pylori*	- Induce intense chronic inflammation in gastric epithelium. Directly induce gene mutation and protein modulation of hosT-cells
Salt-preserved foods	- Alter the viscosity of gastric mucous, making it more sensitive to other carcinogens. - Increase S-phase cell numbers which are susceptible to mutation. - Enhance *H. pylori* colonization by increase surface mucous mucin.
Dietary nitrites	- Convert to nitrite and N-nitroso compounds which are carcinogens. Damage gastric mucosa and cause DNA mutations by release of free radicals.
Gastric surgery	- Chronic inflammation from bile reflux into the stomach and decreases acid production in the stomach.
Alcohol	- Facilitates other carcinogens by acting as solvents. - Contains aldehyde, which is an animal carcinogen - Hard liquor may contain N-nitroso compounds.
Smoking	- Contains more than 60 human carcinogens, including N-nitroso compounds.
Obesity	- Alter insulin and insulin growth factors signaling. - Put the body into a chronic inflammatory state. - Alter sex hormones metabolism to favor tumorigenesis.
Occupational risk	- Physical agents, such as asbestos, may act as co-carcinogens and cause direct irritation to the gastric mucosa. - Fertilization used in agriculture may contain N-nitroso compounds. - Chemical specks of dust can act as a carrier for other carcinogens which damage gastric tissues.

## Data Availability

Not applicable.

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
