# Peer review of "Inflammation and Gastric Cancer"

_diseases, 2022, doi:10.3390/diseases10030035_

Round 1
Reviewer 1 Report
In the present study, Jaroenlapnopparat et al. reviewed the relationship between inflammation and gastric cancer microRNAs and highlighted the role of inflammatory responses in different pathophysiological events during gastric carcinogenesis. There are several kinds of literature on this issue. Searching the PubMed database yielded several reviews on the effects of inflammation on gastric cancer development and progression. This review will cover this field. It is a well-written and thorough review. However, further explanations and drafting are required to better understand the topic. Reading the manuscript resulted in the following comments, and the manuscript would benefit from significant proofreading and editing.
Comments:
Please explain in more detail the significance of the current review in both abstract and text and highlights the novelty of the research.
The authors made a mistake in uploading the figures. Without the figures, this review article is invaluable. I only had access to a figure as a supplementary, which is not explained in the text of the article.
The manuscript does not provide the systematic mechanisms of inflammatory responses in gastric cancer development. It seems been mentioned, but it should be extensively discussed, and a table or figure is preferred to be added to list or summarize all underlying mechanisms.
The full term for which an abbreviation stands should precede its first use in the text.
In the discussion section, the novelty of the study should be addressed significantly.
Some references are extremely out of date and must be replaced by more recent and relevant original studies.
Since the selected topic is enormous, it would be difficult to generate a comprehensive review about the area, however, a better organization and structure of the different sections can help to incorporate more relevant information into the topic without considerably increasing the length of the manuscript.
The manuscript has no conclusion section. Although this review consists of a good and balanced synthesis that covers the general aspects of inflammation and gastric cancer, I recommend including the conclusions section.
Although the language of the review is adequate and easy to follow, some sentences would benefit from editing.
Author Response
Thank you very much for considering our manuscript entitled “Inflammation and Gastric Cancer” for publication in the “Diseases”. We also thank the Editorial Board and the Reviewers for their crucial contributions.
We have reviewed all the valuable comments made by the Reviewers and revised the manuscript accordingly. Below please find the list of the corrections. Please do not hesitate to make any further changes according to the requirements of the journal.
Yours faithfully
Sahin Coban, MD

Reviewer 2 Report
Dear Authors,
I thank You for giving me the opportunity to read this Your manuscript, submitted for publication in Diseases.
I read it with interest.
Your review article is well-written, well organized and comprehensively described, and has several point of interest to readers.
I have no comment or suggestion.
Author Response
Thank you very much for considering our manuscript entitled “Inflammation and Gastric Cancer” for publication in the “Diseases”. We also thank the Editorial Board and your crucial comments.
Reviewer 3 Report
I appreciate the great efforts that the authors have made. I believe this review article should be published.
Author Response
Thank you very much for considering our manuscript entitled “Inflammation and Gastric Cancer” for publication in the “Diseases”. We also thank the Editorial Board and for your crucial comments.
Round 2
Reviewer 1 Report
The authors in the manuscript entitled “Inflammation and Gastric Cancer” aimed at determining the relationship between inflammation and gastric cancer microRNAs and highlighted the role of inflammatory responses in different pathophysiological events during gastric carcinogenesis. I think the authors have successfully revised their manuscript and this paper is mostly easy to follow. In the revised version of the manuscript, the authors carefully addressed my earlier concerns points accordingly. This version of the manuscript now seems to be well-written and should provide valuable findings. Thus I recommend the paper be published.